# In-hospital versus after-discharge complete revascularization in patients with ST segment elevation myocardial infarction and multivessel disease. REVIVA-ST trial

Eva Rumiz[1,2,3]*, Ernesto Valero[2,4,5], Carmen Fernandez[1], Juan Vicente Vilar[1,2], Mauricio Pellicer[4], Andres Cubillos[1], Alberto Berenguer[1], Lorenzo Facila[1], Joan Vaño[6], Julio Nuñez[4,5]

1 Cardiology Department, Consorcio Hospital General Universitario, Valencia, Spain, 2 Cardiology Department, Hospital QuironSalud, Valencia, Spain, 3 Universitat Jaume I, Castellón de la Plana, Spain, 4 Cardiology Department, Hospital Clínico Universitario, INCLIVA, Universitat de València, Valencia, Spain, 5 CIBER Cardiovascular, Madrid, Spain, 6 Hospital Lluis Alcanyis, Xativa, Valencia, Spain

* evarumizgonzalez@gmail.com

**Data Availability Statement:** All relevant data are within the manuscript and its Supporting Information files.

## Abstract

### Introduction

Complete revascularization (CR) in patients with ST-segment elevation myocardial infarction (STEMI) and multivessel disease (MVD), is associated with a reduction in major adverse cardiovascular events (MACE). However, there is uncertainty about whether non-culprit-lesion revascularization should be performed, during index hospitalization or delayed, especially regarding health care resources utilization. In this study, we aimed to evaluate the impact of in-hospital nonculprit-lesion revascularization vs. delayed (after discharge) revascularization on the length of index hospitalization.

### Methods

In this single-center study, we randomly assigned patients with STEMI and MVD who underwent successful culprit-lesion PCI to a strategy of either CR during in-hospital admission or a delayed CR after discharge. The first primary endpoint was the length of hospital stay. The second endpoint was the composite of cardiovascular death, myocardial infarction or ischemia-driven revascularization at 12 months (MACE).

### Results

From January 2018 to December 2022, we enrolled 258 patients (131 allocated to CR during in-hospital admission and 127 to an after-discharge CR). We found a significant reduction in the length of hospital stay in those assigned to after-discharge CR strategy [4 days (3–5) versus 7 days (5–9); p = 0.001]. At 12-month of follow-up, no differences were found in the occurrence of MACE, 7 (5.34%) patients in in-hospital CR and 4 (3.15%) in after-discharge CR strategy; (hazard ratio, 0.59; 95% confidence interval, 0.17 to 2.02; p = 0.397).

**Funding:** The authors received no specific funding for this work.

**Competing interests:** The authors have declared that no competing interests exist.

## Conclusions

In STEMI patients with MVD, an after-discharge CR strategy reduces the length of index hospitalization without an increased risk of MACE after 12 months of follow-up.

## Trial registration

**ClinicalTrials.gov number:** NCT04743154.

## Introduction

Primary percutaneous coronary intervention (PCI) is the treatment of choice for patients with ST-segment elevation myocardial infarction (STEMI), as it is demonstrated to reduce the risk of cardiovascular death or myocardial infarction (MI) [1]. About 50% of STEMI patients have multivessel coronary artery disease (MVD), fact that implies a worse prognosis when compared with patients with a single-vessel disease [2,3]. In this particular scenario, a complete revascularization (CR) strategy of the nonculprit lesions has been associated with a reduction in major adverse cardiovascular events (MACE) [4–8]. However, the optimal timing to perform this "protective" PCI remains uncertain. The 2017 European Society of Cardiology (ESC) STEMI Guidelines recommended a CR strategy preferably during index admission before discharge [9]. The main rationale for this recommendation was due to the fact that all trials available until then had performed MVD PCI within that specific time frame. However, hypothetically, this non-delayed strategy may increase the risk of side effects (such as contrast induced nephropathy or bleeding) and logistic constrains that ultimately increase the length of index hospitalization. Recently, the COMPLETE trial, demonstrated a 3-year reduction in the combined endpoint of death or MI with staged PCI (performed within 45 days of STEMI) of the non-infarct related artery compared with culprit-vessel only PCI [8]. These benefits were consistent, irrespective of the timing of the non-infarct artery PCI, whether it was performed during admission or after discharge, but always within the first 45 days. Therefore, the recent 2023 ESC Guidelines for the management of acute coronary syndromes, recommend a CR either during the index PCI procedure or within 45 days (class of recommendation I, level of evidence A) [10]. Since the optimal timing of revascularization (immediate during the index procedure vs. staged within 45 days) has not yet been evaluated in adequate randomized clinical trials (RCTs), no recommendation has been endorsed in this sense in the guidelines. Both of these strategies have pros and cons [11], and therefore, the optimal timing for CR still today remains a subject of debate.

Along with these lines of thoughts, we speculate that a delayed PCI of the nonculprit lesions may tanslate into a shorter index hospitalization length without penalization regarding adverse clinical events. Thus, in this study, we aimed to examine the effect of both CR strategies on the length of stay of index hospitalization in STEMI patients with MVD and secondarily their MACE events.

## Methods

### Overall study design

The REVIVA-ST trial is an investigator-initiated, single-center, open-label RCT that evaluated a strategy of in-hospital CR as compared with a strategy of after-discharge CR in STEMI patients with MVD. We conducted this trial in a third-level university hospital with the

participation of two second-level hospitals without a cardiac catheterization laboratory but with cardiology in-patient wards.

The eligible patients signed informed consent before being randomized 1:1 ratio to receive a in-hospital or after-discharge CR of the nonculprit lesions. A centralized automated system was used to randomly assign patients to both strategies.

All participants received a unique and permanent number that enables identify individual participants during and after data collection.

The investigation conforms to the principles outlined in the Declaration of Helsinki and Good Clinical Practice of the International Conference on Harmonization. The study protocol was approved by Comité Ético de Investigación Clínica (CEIC) del Consorcio Hospital General Universitario de Valencia before the study was initiated, and it was registered at http://clinicaltrials.gov (NCT04743154). The registration in clinical trials.gov was delayed due to administrative issues, since it was not a mandatory requirment in our country at the beginning of the study.

The authors confirm that all ongoing and related trials for this intervention are registered.

## Study population

All adult patients ($\geq$ 18 years) with STEMI who underwent a successful primary PCI and angiographic evidence of MVD were candidates for enrollment. Successful PCI of the culprit lesion was defined as having a Thrombolysis in Myocardial Infarction (TIMI) score of at least 2 and a residual stenosis measure of less than 30% for the culprit lesion. MVD was defined as the presence of at least one significant lesion in a nonculprit coronary artery, which was amenable to successful treatment with PCI and located in a vessel with a diameter of at least 2.0 mm. Angiographically significant lesions were deemed when the vessel diameters presented at least a 70% stenosis or between $\geq$50% and <70% in proximal segments, on visual estimation. The nonculprit lesion was never stented as part of the index culprit-lesion PCI. The main exclusion criteria were patients with cardiogenic shock at admission, left main coronary disease ($\geq$50% diameter stenosis), previous coronary artery bypass grafting (CABG), presence of a chronic total occlusion, and referral for CABG.

Randomization was performed during early hospitalization (no later than 48 hours) after the index PCI. Patients who were randomly assigned to in-hospital CR were allocated to receive a staged PCI during admission at least 24 hours after index PCI, of all suitable nonculprit lesions. Patients who were randomly assigned to after discharge CR, were assured to have a staged PCI of all suitable nonculprit lesions after discharge no later than 6 weeks.

Revascularization guided by fractional flow reserve (FFR) was encouraged in stenosis less than 70%, by means of a PressureWire® (Abbott medical, Belgium). An FFR value of 0.80 or less was considered to be clinically significant, recommending to perform PCI on the corresponding lesion.

Clinical and laboratory data were registered from medical records. All patients were discharged with optimal medical treatment according to the current recommendations for clinical practice [9,10].

## Endpoints and follow-up

The primary endpoint was to evaluate the differences in length of hospital stay between treatment strategies. The secondary endpoints were: the composite of cardiovascular death, MI, or ischemia-driven revascularization during 12 months follow-up; as well as, to study the influence of the timing of revascularization on fractional flow reserve (FFR) measurements of intermediate nonculprit lesions located at proximal coronary segments.

The length of hospital stay was recorded from the day of admission to the day of hospital discharge. The need for a hospital admission longer than 24 hours after elective PCI was also recorded as hospital stay in the group of patients assigned to the after-discharge strategy. MI was defined according to the fourth universal definition and was subclassified according to type [12]. For defining ischemia-driven revascularization, we included all PCI or CABG occurring after randomization and justified by recurrent symptoms or objective evidence of significant ischemia on provocative testing. Follow-up was conducted during outpatient clinic visits scheduled at 30 days, 6 months and 12 months after primary revascularization, and also by means of hospital electronic database. All events were identified and ascertained by consensus of two independent cardiologists that were blinded to treatment allocation and medical history of the participants.

The impact of the timing of revascularization on the assessment of FFR was assessed through the analysis of mean FFR and the percentage of guided-FFR revascularization between both groups.

### Statistical analysis

Discrete data are expressed as frequency and percentages, and continuous data are presented as means (±SD) or medians and interquartile ranges as appropriate.

All patients who underwent randomization were included in the analysis according to the treatment group to which they were assigned, regardless of the CR strategy they actually received (intention-to-treat principle). Because the length of stay did not meet the criteria for a parametric distribution (assessed by Shapiro-Wilk test), the primary endpoint (length of stay) between treatment arms was analyzed by the Mann-Whitney test. The secondary outcome (time to 12-month MACE) was analyzed by Kaplan Meier analysis, and differences were quantified by the log-rank test. Estimates of risk were also assessed by using Cox regression analysis, and differences expressed as unadjusted hazard ratios (HR) and 95% confidence interval (CI). Proportional-hazard assumption of the survival model was verified by using Schoenfeld residuals (p-value: 0.703). A 2-sided p-value of 0.005 was considered significant for all the analyses. All analyses were performed using STATA 17 (Stata Statistical Software, College Station, TX).

**Sample size.** We calculated that a sample size of 250 patients (125 patients per group) would give the trial a power of at least 90% (at a two-sided alpha of 5%) to detect a reduction in two days of hospitalization in in-hospital CR strategy when compared with the delayed CR strategy. We assumed 20% of the participants would withdraw or be lost to follow-up. This estimate was based on the results of previous studies [13,14]. The software used for the sample size calculation was "xsampsi" from Stata 13.1.

## Results

From January 2018 to December 2021, a total of 258 patients were enrolled and underwent randomization, 131 were assigned to CR during in-hospital admission and 127 were assigned to a staged CR after discharge. A detailed flow chart of patients is provided in Fig 1.

There were 3 cross-over during the trial. Three patients of in-hospital CR group crossed over to the group of after-discharge CR, 1 patient for severe thrombocytopenia induced by heparin, and 2 due to acute renal failure.

The baseline characteristics were similar in the two treatment groups (Table 1). The median age of the study population was 61.5 years (interquartile range, 55–71), and 89.9% were male. The procedural characteristics were also well-balanced (Table 2). The right coronary artery was the culprit vessel in 46.5% of patients, and most of them (93.4%) were in Killip class I at admission. The mean number of residual diseased vessels pending revascularization was 1.3 ± 0.5, in both groups. The residual Syntax score was also similar between the two strategies.

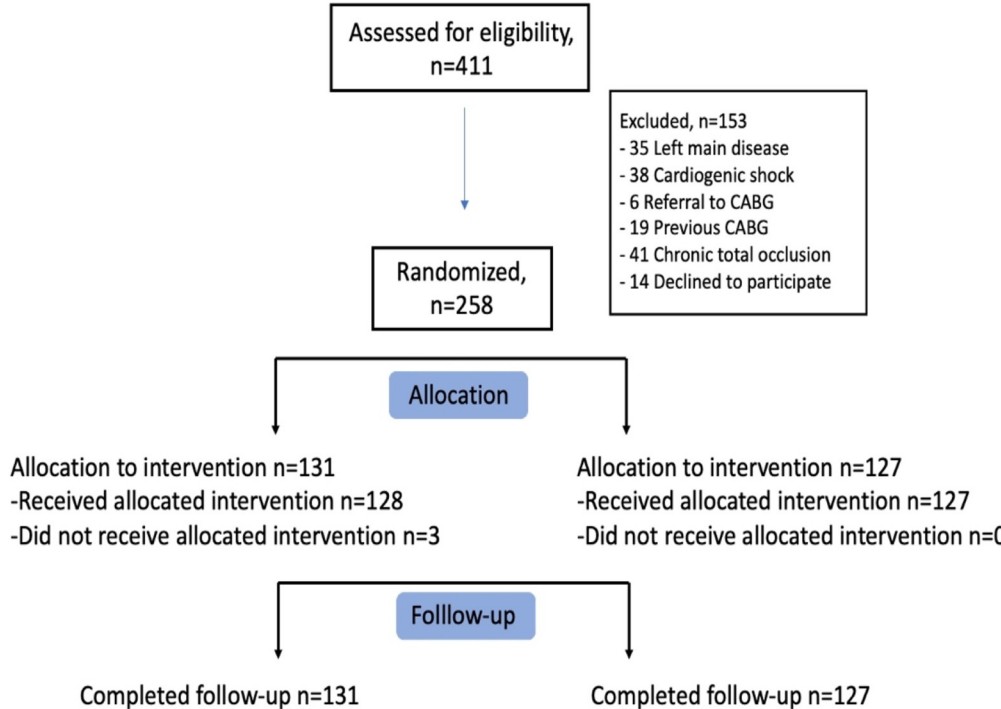

**Fig 1. Study flow chart.** CR, complete revascularization; CABG, coronary artery bypass grafting.

The left anterior descending coronary artery was the nonculprit vessel more affected in both groups. The median time until CR was 2 (1–3) and 22 days (18–26), for in-hospital and delayed CR strategies, respectively. No differences were observed between the two group strategies in terms of peri-procedural major bleeding (BARC ≥ 3) or vascular access complications.

**Table 1. Baseline characteristics of the patients stratified by randomization arm.**

| | In-hospital CR (n = 131) | After-discharge CR (n = 127) | p-value |
|---|---|---|---|
| Age (years) | 62 (54–71) | 61 (55–72) | 0.975 |
| Male sex, n (%) | 118 (90.1) | 114 (89.8) | 0.934 |
| Hypertension, n (%) | 76 (58) | 82 (64.6) | 0.280 |
| Diabetes, n (%) | 42 (32.1) | 36 (28.3) | 0.516 |
| Dyslipidemia, n (%) | 79 (60.3) | 68 (53.5) | 0.273 |
| Chronic kidney disease[a], n (%) | 9 (6.9) | 7 (5.5) | 0.651 |
| Obesity, n (%) | 17 (13) | 13 (10.2) | 0.492 |
| Current smoker, n (%) | 71 (54.2) | 60 (47.2) | 0.053 |
| Previous IHD, n (%) | 13 (9.9) | 6 (4.7) | 0.110 |
| Previous stroke, n (%) | 7 (5.3) | 3 (2.4) | 0.215 |
| Anterior STEMI, n (%) | 47 (35.9) | 50 (39.4) | 0.504 |
| Killip class I, n (%) | 119 (90.8) | 122 (96.1) | 0.154 |
| Creatinine at baseline, mg/dl | 1.5 ± 0.55 | 0.98 ± 0.75 | 0.862 |

Data are given as n (%), or median (interquartile range).

CR: Complete revascularization; IHD: Ischemic heart disease; STEMI: ST-segment elevation myocardial infarction.

[a] Estimated glomerular filtration rate <60ml/min/1.73m$^2$ by Cockcroft-Gault equation.

**Table 2. Procedural and at discharge characteristics.**

| | In-hospital CR (n = 131) | After-discharge CR (n = 127) | p-value |
|---|---|---|---|
| Radial Access, n (%) | 123 (93.9) | 115 (91.3) | 0.133 |
| Volume of contrast used, ml | 110 (90–115) | 105 (72–108) | 0.746 |
| No. Stents, | 1.28 ± 0.61 | 1.27 ± 0.52 | 0.572 |
| Predilatation, n (%) | 114 (87) | 107 (84.2) | 0.505 |
| Location of culprit lesion, n (%) | | | 0.650 |
| Left anterior descending | 49 (37.4) | 50 (39.4) | |
| Circumflex | 24 (18.3) | 15 (11.8) | |
| Right coronary | 58 (44.3) | 62 (48.8) | |
| Segment of culprit lesion, n (%) | | | 0.147 |
| Proximal | 59 (45) | 51 (40.2) | |
| Middle | 46 (35.1) | 54 (42.5) | |
| Distal | 24 (18.3) | 22 (17.3) | |
| Residual diseased vessels, n (%) | | | 0.543 |
| 1 | 99 (75.6) | 91 (71.6) | |
| ≥2 | 32 (24.4) | 36 (28.3) | |
| No. of residual diseased vessels | 1.3 ± 0.5 | 1.3 ± 0.5 | 0.991 |
| Location of nonculprit lesion, n (%) | | | |
| *Left anterior descending* | | | |
| Proximal | 23 (17.7) | 14 (11) | 0.234 |
| Middle | 44 (33.6) | 46 (36.2) | 0.657 |
| *Circumflex artery* | | | |
| Proximal | 22 (16.8) | 27 (21.2) | 0.354 |
| Distal | 29 (22.1) | 33 (26) | 0.470 |
| *Right Coronary artery* | | | |
| Proximal | 11 (8.4) | 20 (15.9) | 0.066 |
| Middle | 29 (22.1) | 23 (18.1) | 0.420 |
| Residual Syntax score | 5 (2–9) | 4 (2–8) | 0.379 |
| LVEF (%) | 55 (46–60) | 55 (45–60) | 0.516 |
| P2Y$_{12}$ inhibitor at discharge, n (%) | | | 0.544 |
| Clopidogrel | 50 (43.1) | 43 (39.1) | |
| Prasugrel | 32 (27.6) | 38 (34.5) | |
| Ticagrelor | 34 (29.3) | 29 (26.4) | |
| Hospital Length (days) | | | |
| CCU | 2 (2–3) | 2 (2–3) | 0.990 |
| Total | 7 (5–9) | 4 (3–5) | <0.001 |
| Creatinine at 12-month follow-up, mg/dl | 1.48 ± 0.43 | 0.96 ± 0.61 | 0.634 |
| Peri-procedural major bleeding, n (%) | 2 (1.53) | 0 (0) | 0.161 |
| Vascular access complication, n (%) | 1 (0.76) | 0 (0) | 0.320 |

Data are given as n (%), median (interquartile range), or mean ± SD.

CCU: Coronary care unit; CR: Complete revascularization; No: Number; LVEF: Left ventricle ejection fraction.

## Primary endpoint

Clinical outcomes are summarized in Table 3. The median length of hospital stay (p25%-p75%) was significantly higher in the in-hospital CR strategy, when compared with the after-discharge CR group [7 days (5–9) versus 4 days (3–5); p<0.001]. There were no differences between both strategies in terms of length of stay in the median coronary care unit (CCU)

**Table 3. Primary and secondary outcomes.**

| | In-hospital CR (n = 131) | After-discharge CR (n = 127) | Hazard ratio (95% CI) | p-value |
|---|---|---|---|---|
| Primary outcome<br>Total hospital stay (days) | 7 (5–9) | 4 (3–5) | | <0.001 |
| Secondary outcome<br>All-cause death, myocardial infarction or ischemia-driven revascularization at 12 months, n (%) | 7 (5.34) | 4 (3.15) | 0.59 (0.17–2.02) | 0.397 |
| All cause death | 4 (3.05) | 2 (1.57) | | 0.431 |
| Cardiovascular death | 2 (1.53) | 2 (1.57) | | 0.626 |
| Myocardial infraction | 3 (2.29) | 2 (1.57) | | 0.677 |
| Ischemia-driven revascularization | 3 (2.29) | 2 (1.57) | | 0.677 |

Data are given as n (%) or median (interquartile range).

CI: Confidence interval; CR: Complete revascularization.

[2 days (2–3) vs 2 days (2–3), p = 0.631], respectively. The median length of stay after a non-culprit revascularization procedure in the after-discharge strategy was inferior to 24 hours in all patients [0.38 days (0.25–0.81)].

## Secondary endpoint

At 12-month of follow-up, the secondary endpoint occurred in 7 of 131 patients (5.34%) in the in-hospital CR group and in 4 of 127 (3.15%) in the after-discharge CR strategy. Kaplan-Meier curves showed no significant differences along the entire follow-up (Fig 2). Cox regression

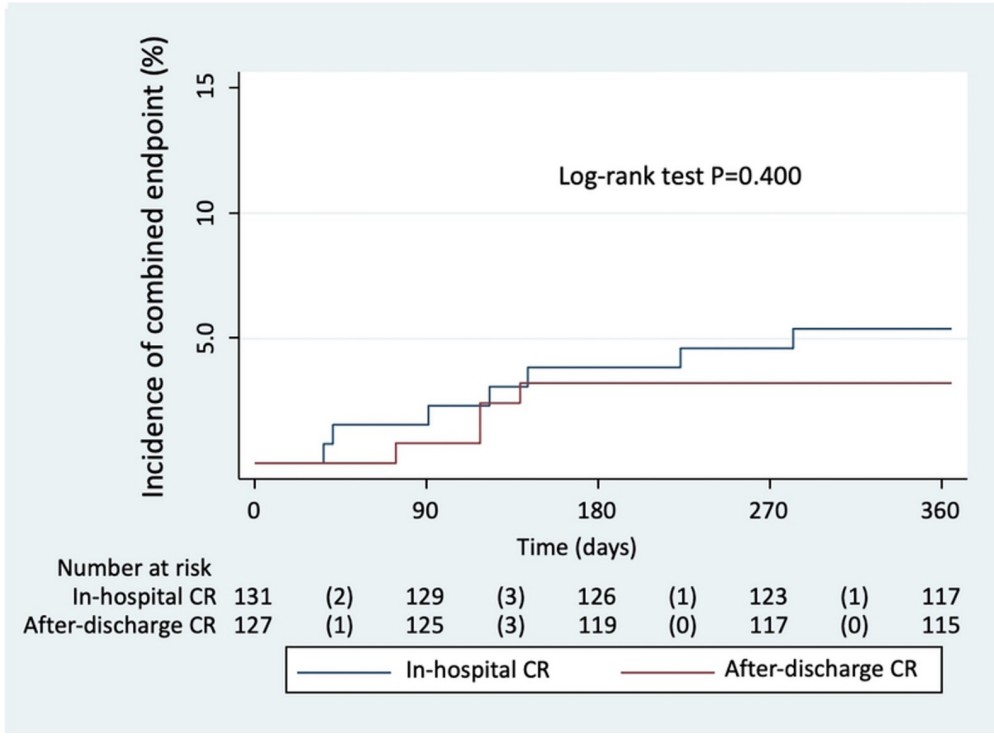

**Fig 2. Kaplan-Meier estimate of the combined secondary endpoint at 12 months.** CR, complete revascularization.

analysis also revealed a neutral effect of delayed revascularization on the risk of 12-month MACE [HR, 0.59; 95% CI, 0.17–2.03; p = 0.407]. No adverse clinical events occurred in any of the different strategies until the nonculprit lesion revascularization was performed.

FFR-guided revascularization was performed in 65 lesions, 32 in the in-hospital CR group and 33 in the after-discharge CR strategy. No differences in the mean FFR were observed between both groups, 0.91 ± 0.12 and 0.88 ± 0.10; (p = 0.724), respectively. Hemodynamically significant FFR (defined as ≤0.8) were obtained in 10 lesions (31.2%) of the in-hospital group and 12 (36%) of the after-discharge strategy (p = 0.610).

## Discussion

The main finding of the current trial is that in STEMI patients with MVD, a delayed CR strategy of the nonculprit lesions (with a median of 22 days after STEMI), translates into a significant reduction in the length of hospital stay without an increase in the risk of 12-month MACE.

### Complete revascularization strategies and hospital length

To the best of our knowledge, this is the first RCT to evaluate the impact of different revascularization strategies on the length of hospital stay in patients with STEMI and MVD. Multiple studies have evaluated the efficacy of a CR in STEMI patients with MVD, however, even today, the optimal time to perform it still remains a matter of debate [15]. In this regard, our study demonstrates a significant benefit in terms of reduction in hospital stay achieved by the after-discharged CR approach when compared with a CR performed during index admission.

Among the benefits of an early hospital discharge would be an increased efficiency of health care resource use and the reduction of hospital expenditures. Moreover, this strategy is welcomed by patients for obvious reasons, since it shortens hospital stay, and also allows clinicians to rapidly enroll patients in a cardiac rehabilitation programme. In this sense, the current ESC Guidelines suggest that an early discharge (within 48–72 h) would be considered appropriate in selected low-risk patients [9,10]. By contrast, the previous ESC guidelines recommended routine revascularization of nonculprit lesions in STEMI patients with MVD before hospital discharge [9]. However, our results show that this revascularization approach, during the index admission, carries an increase in hospital length. Furthermore, our study underlines the safety of a staged approach in these patients, probably related to the efficacy of modern drug-eluting stents (DES), and in contrast to the trials included in the previous ESC Guidelines where a significant part of the patients were no treated with DES [9]. In this regard, our findings in terms of safety are in line with a recent Meta-analysis published by Panuccio et al, demonstrating the safety of the use of DES in terms of clinical outcomes in patients with STEMI and MVD [16].

Although an immediate, during primary PCI, nonculprit PCI strategy has not been explored in our work, we believe that it would not lead to a further reduction in the length of stay. Recently, the Multivessel Immediate versus Staged Revascularization in Acute Myocardial Infarction (MULTISTARS AMI) trial, demonstrated that immediate multivessel PCI at the time of primary PCI was noninferior to staged multivessel PCI in patients with STEMI and MVD, reporting a median hospital stay of 4 days (3–6) in the immediate CR strategy, similar to the data obtained in the delayed CR group of our work [17]. Likewise, Politi et al observed no differences in this setting between STEMI patients revascularized during primary PCI or in a staged procedure [18]. By contrast, the BIOVASC trial showed a reduction in hospital stay in patients with acute coronary syndrome (ACS) who underwent an inmediate CR when compared with patients with a staged CR approach [19]. However, in this study only 40% of patients presented with STEMI and the staged PCI could be performed within 6 weeks after

the index PCI, both during the hospitalization or after discharge. Moreover, we do not actually know whether the increase in hospital stay is a consequence of a longer index admission or secondary to the deferred procedure.

Furthermore, we observed no differences in terms of stay in the CCU between both strategies. The median stay was 2 days (IQR 2–3), which is in agreement with the ESC guidelines that recommend at least 24 hours in the CCU after successful reperfusion therapy in an uncomplicated STEMI [9,10].

## Complete revascularization strategies and risk of MACE

The vast majority of RCTs concerning MVD in STEMI patients show a reduction in terms of MACE, observing a significant benefit of a CR approach when compared with a culprit-vessel only PCI strategy [4–8]. The timing in which nonculprit-lesions PCI were performed in these RCTs differs from immediate to staged PCI, being more often undertook during index admission. The current guidelines recommendations on revascularization of nonculprit lesions in STEMI [9,10], are based on the data from RCTs that evaluate the benefit of a CR over culprit-vessel only PCI. However, as far as we know, actually exists scarce scientific evidence that focuses on the evaluation of the optimal-timing of the different CR strategies in STEMI patients with MVD. The BIOVASC trial [19] showed that an immediate CR was non-inferior to a staged CR for the combined endpoint (all cause mortality, MI, unplanned ischaemia-driven revascularization or cerebrovascular event) at 12 months of follow-up. However, this trial was performed in a cohort of patients with ACS, where 60% were non-STEMI patients. In our study, we did not observe any significant relationship between the timing of revascularization and the rates of MACE at 12 months. It is important to highlight that no events occurred in any strategy until CR was performed. By contrast, the BIOVASC trial observed an excess of MI in the staged strategy mainly driven by events occurred until de CR was performed [19]. Accordingly, the MULTISTARS trial also presented an increase in 12-month MACE, mostly due to nonfatal MI and unplanned ischemia-driven revascularization, in the staged CR strategy. It is important to highlight that, no apparent differences in the percentage of patients with spontaneous MI were observed between the trial groups. Furthermore, the difference observed was predominantly due to an elevated rate of procedure-related MI, which is a really difficult MACE to quantify in an immediate strategy.

In this sense, our findings are in line with the recently published COMPLETE trial, that evaluated the rate of cardiac death or MI in culprit-vessel only PCI versus a staged nonculprit-lesions PCI. The authors performed a randomization stratified according to the intended timing of nonculprit-lesion PCI. This is up to date, the only RCT that demonstrates a clear benefit of CR, regardless of whether nonculprit-lesion PCI was performed during the index hospitalization or delayed several weeks after discharge[8].

Finally, we also explored the possible influence of the timing of revascularization on the functional assessment of nonculprit lesions by FFR analysis. Van der Hoeven et al, showed an increase of nonculprit FFR values in the acute setting of STEMI as compared to 1-month FFR values [20]. This findings could be explained by a blunted adenosine responsiveness due to a decrease in the sensitivity of the purinergic adenosine receptors even in the remote myocardium at the acute moment. Although in a small number of patients and in a subacute setting, we did not observe any differences in the rate of FFR-guided revascularization between the two groups.

## Study limitations

Some limitations need to be acknowledged. First, this is a single-center RCT in which all PCI were centralized in the same hospital, therefore hidden bias might be operating. We should

bear in mind that the celerity with which the second revascularization is performed could influence the results. Second, other CR strategies as immediate, during primary PCI revascularization strategy of the nonculprit lesions were not evaluated.

## Conclusion

In patients with STEMI and MVD, an after-discharge CR strategy reduces the length of hospitalization without an increased risk of cardiovascular death, MI or ischemia driven revascularization at 12 months of follow-up.

## Supporting information

**S1 Checklist.**
(DOCX)

**S1 Graphical abstract. CV, cardiovascular; MVD, multivessel disease; PCI, percutaneous coronary intervention; STEMI, ST-segment elevation myocardial infarction.**
(JPG)

**S1 File.**
(DOCX)

**S2 File.**
(XLSX)

## Author Contributions

**Conceptualization:** Eva Rumiz.

**Data curation:** Eva Rumiz, Carmen Fernandez, Juan Vicente Vilar, Andres Cubillos, Alberto Berenguer, Joan Vaño, Julio Nuñez.

**Formal analysis:** Eva Rumiz, Julio Nuñez.

**Investigation:** Eva Rumiz, Ernesto Valero, Carmen Fernandez, Juan Vicente Vilar, Mauricio Pellicer, Andres Cubillos, Alberto Berenguer, Lorenzo Facila, Joan Vaño, Julio Nuñez.

**Methodology:** Eva Rumiz, Ernesto Valero, Mauricio Pellicer, Andres Cubillos, Alberto Berenguer, Lorenzo Facila, Joan Vaño, Julio Nuñez.

**Project administration:** Eva Rumiz, Julio Nuñez.

**Supervision:** Eva Rumiz, Juan Vicente Vilar, Alberto Berenguer, Julio Nuñez.

**Validation:** Eva Rumiz, Juan Vicente Vilar, Mauricio Pellicer, Andres Cubillos, Alberto Berenguer, Lorenzo Facila, Joan Vaño, Julio Nuñez.

**Visualization:** Eva Rumiz, Carmen Fernandez, Mauricio Pellicer, Andres Cubillos, Alberto Berenguer, Lorenzo Facila, Joan Vaño, Julio Nuñez.

**Writing – original draft:** Eva Rumiz, Ernesto Valero, Julio Nuñez.

**Writing – review & editing:** Eva Rumiz, Ernesto Valero, Julio Nuñez.

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
