## [Decision Letter · Decision Letter 0]

16 Aug 2023

PONE-D-23-18808In-hospital versus after-discharge complete revascularization in patients with ST segment elevation myocardial infarction and multivessel disease. REVIVA-ST trial.PLOS ONE

Dear Dr. Rumiz,

Thank you for submitting your manuscript to PLOS ONE. After careful consideration, we feel that it has merit but does not fully meet PLOS ONE’s publication criteria as it currently stands. Therefore, we invite you to submit a revised version of the manuscript that addresses the points raised during the review process.

Please carefully addreess all comments by the expert Reviewers.

We look forward to receiving your revised manuscript.

Kind regards,

Salvatore De Rosa

Academic Editor

PLOS ONE

2. We note that you have selected “Clinical Trial” as your article type. PLOS ONE requires that all clinical trials are registered in an appropriate registry (the WHO list of approved registries is at https://www.who.int/clinical-trials-registry-platform/network/primary-registries"" https://www.who.int/clinical-trials-registry-platform/network/primary-registries" https://www.who.int/clinical-trials-registry-platform/network/primary-registries and more information on trial registration is at http://www.icmje.org/about-icmje/faqs/clinical-trials-registration/). Please state the name of the registry and the registration number (e.g. ISRCTN or ClinicalTrials.gov) in the submission data and on the title page of your manuscript. a) Please provide the complete date range for participant recruitment and follow-up in the methods section of your manuscript. b) If you have not yet registered your trial in an appropriate registry, we now require you to do so and will need confirmation of the trial registry number before we can pass your paper to the next stage of review. Please include in the Methods section of your paper your reasons for not registering this study before enrolment of participants started. Please confirm that all related trials are registered by stating: “The authors confirm that all ongoing and related trials for this drug/intervention are registered”. Please see http://journals.plos.org/plosone/s/submission-guidelines#loc-clinical-trials for our policies on clinical trials.

Reviewers' comments:

Reviewer's Responses to Questions

**Comments to the Author**

1. Is the manuscript technically sound, and do the data support the conclusions?

Reviewer #1: Yes

Reviewer #2: Yes

2. Has the statistical analysis been performed appropriately and rigorously? 

Reviewer #1: Yes

Reviewer #2: Yes

3. Have the authors made all data underlying the findings in their manuscript fully available?

Reviewer #1: Yes

Reviewer #2: No

4. Is the manuscript presented in an intelligible fashion and written in standard English?

Reviewer #1: Yes

Reviewer #2: Yes

5. Review Comments to the Author

Reviewer #1: In this trial, the colleagues compared two different strategies of complete coronary revascularization in ACS STEMI patients, showing higher hospital stay in in hospital admission complete revascularization strategy and no difference in term of MACE.

The topic of the manuscript is interesting, however there are some issues to solve and revise:

1) There are too many typos in the text (some examples: trategies in the last sentence of introduction section, PressueWire, also a lot of times the authors write “de” instead of “the”). Please check all the typos and revise.

2)The endpoints description should be written more fluently and not as a list. Please revise. Moreover, after double point (: ) the authors should not write in uppercase. Please revise.

3) The authors report no difference in intensive care unit stay between the two groups. Please specify if the intensive care stay in the staged revascularization group is related to first or second hospital admission ( and if it is related to the second admission, please clarify why elective patients treated for elective PCI should go in coronary intensive care unit).

4)One of the endpoints analyzed in several trials regarding this topic was cardiovascular death. If the authors have those data, they should report them in the manuscript.

5) The authors should underline the safety of a staged approach nowadays, probably related to the efficacy of modern drug eluting stents. Most of the trials included also in the recent guidelines sometimes included patients that were not treated with DES. At this regard I suggest citing the following work that underlined this concept and performed an analysis on the effect of DES in clinical outcomes in patients with STEMI and MVD:

-Panuccio G, Salerno N, Rosa SD, Torella D. Timing of Complete Revascularization in Patients with STEMI and Multivessel Disease: A Systematic Review and Meta-Analysis. RCM 2023;24. doi: 10.31083/j.rcm2402058

Reviewer #2: Please see the following comments and questions where the manuscript is quoted followed by the question:

Introduction

‘major cardiovascular adverse events (MACE)’ – please revise to standard work sequence to match the acronym

‘…(such as increasing the stress of the patients,…’ - I would consider removing this if you do not have references to support. In my opinion some patients may be stressed to wait in hospital while others may to stressed to have lesions left unmanaged at discharged. Accordingly, best to just remove this statement.

‘Thus, in this study, we aimed to examine the effect of both CR trategies on the length of stay of index hospitalization in STEMI patients with MVD.’ Since your hypothesis is also that this will be completed safely, I would consider revising to read something like this – if you agree “”Thus, in this study, we aimed to examine the effect of both CR strategies on the length of stay of index hospitalization in STEMI patients with MVD and secondarily their MACE events.””

Methods

‘For defining ischemia-driven revascularization, we included all PCI or CABG occurring after the baseline procedure and justified by recurrent symptoms or objective evidence of significant ischemia on provocative testing.’ Please clarify this statement. I assume that it means all revascularization after randomization and not after the index procedure? It would be atypical to count events prior to randomization.

Results

Abstract ‘At 12-month of follow-up, no differences were found in the occurrence of MACE, 7 (5.34%) patients in in-hospital CR and 4 (3.15%) in after-discharge CR strategy; (hazard ratio, 0.59; 95% confidence interval, 0.17 to 2.02; p=0.397).’

Table 10 (7.63) vs. 6 (4.72) 0.59 (0.17- 2.02)’

Results text section ‘At 12-month of follow-up, the secondary endpoint occurred in 7 of 131 patients (5.34%) in the in-hospital CR strategy and in 4 of 127 (3.15%).

The key secondary results appear different in the abstract, the results table and the results text?

Discussion

‘Although an immediate, during primary PCI, nonculprit PCI strategy has not been explored, we believe that it would not lead to a further reduction in the length of stay.’ I would revise this since there are several trials that have explored immediate culprit plus non-culprit PCI in the setting of MVD and STEMI.

Additional discussion comments – the discussion is a bit repetitive that the current work is novel. Although I agree it is important and novel – this does not need to be stated more that once.

6. PLOS authors have the option to publish the peer review history of their article (what does this mean?). If published, this will include your full peer review and any attached files.

Reviewer #1: No

Reviewer #2: **Yes: **Robert C. Welsh

---

## [Author Response · Author response to Decision Letter 0]

16 Sep 2023

Response to Reviewers

Ref.: PONE-D-23-18808 

We acknowledge to the Editorial Office and Reviewers for their comments and constructive suggestions, which undoubtedly have improved the quality of this manuscript.

Reviewer #1: 

In this trial, the colleagues compared two different strategies of complete coronary revascularization in ACS STEMI patients, showing higher hospital stay in in hospital admission complete revascularization strategy and no difference in term of MACE.

The topic of the manuscript is interesting, however there are some issues to solve and revise:

1) There are too many typos in the text (some examples: trategies in the last sentence of introduction section, PressueWire, also a lot of times the authors write “de” instead of “the”). Please check all the typos and revise.

- Response: Attending the reviewer’s comment, we have checked and revised all the typos present in the manuscript. 

2) The endpoints description should be written more fluently and not as a list. Please revise. Moreover, after double point (: ) the authors should not write in uppercase. Please revise.

- Response: Attending the reviewer’s suggestion, the endpoints description has been written more fluently and not as a list, and after double point we have revised the text. Therefore, the Endpoints and follow-up paragraph (page 5) has been changed: “The primary endpoint was to evaluate the differences in length of hospital stay between treatment strategies. The secondary endpoints were: the composite of cardiovascular death, MI, or ischemia-driven revascularization during 12 months follow-up; as well as, to study the influence of the timing of revascularization on fractional flow reserve (FFR) measurements of intermediate nonculprit lesions located at proximal coronary segments”.

3) The authors report no difference in intensive care unit stay between the two groups. Please specify if the intensive care stay in the staged revascularization group is related to first or second hospital admission ( and if it is related to the second admission, please clarify why elective patients treated for elective PCI should go in coronary intensive care unit).

-Response: Answering the reviewer’s question, the intensive care stay in the staged revascularization group is only related to the first hospital admission (index admission). The after-discharge revascularization was always a programmed-elective procedure through radial artery access with no procedure-related complications, and therefore these patients needed no hospital stay in that second admission, being all discharged home in the same day of the procedure.

4) One of the endpoints analyzed in several trials regarding this topic was cardiovascular death. If the authors have those data, they should report them in the manuscript.

- Response: Attending the reviewer’s suggestion, we have reported the data regarding cardiovascular death in Table 3 (page 9): “Cardiovascular death n(%) – In hospital CR 2 (1.53) – After discharge CR 2 (1.57) – p-value 0.626”. 

5) The authors should underline the safety of a staged approach nowadays, probably related to the efficacy of modern drug eluting stents. Most of the trials included also in the recent guidelines sometimes included patients that were not treated with DES. At this regard I suggest citing the following work that underlined this concept and performed an analysis on the effect of DES in clinical outcomes in patients with STEMI and MVD: -Panuccio G, Salerno N, Rosa SD, Torella D. Timing of Complete Revascularization in Patients with STEMI and Multivessel Disease: A Systematic Review and Meta-Analysis. RCM 2023;24. doi: 10.31083/j.rcm2402058.

- Response: Attending the reviewer’s suggestion, we have added the next paragraph in the Discussion (page 11, paragraph 1, lines 8-13): “Furthermore, our study underlines the safety of a staged approach in these patients, probably related to the efficacy of modern drug-eluting stents (DES), and in contrast to the trials included in the actual ESC Guidelines where a significant part of the patients were no treated with DES. In this regard, our findings in terms of safety are in line with a recent Meta-analysis published by Panuccio et al, demonstrating the safety of the use of DES in terms of clinical outcomes in patients with STEMI and MVD13”. Additionally, we have also included the next reference to the References (page 15, reference 13): “13. Panuccio G, Salerno N, Rosa SD, Torella D. Timing of Complete Revascularization in Patients with STEMI and Multivessel Disease: A Systematic Review and Meta-Analysis. Rev Cardiovasc Med. 2023;24(2), 58. doi: 10.31083/j.rcm2402058”.

 

Reviewer #2: 

Please see the following comments and questions where the manuscript is quoted followed by the question:

Introduction

‘major cardiovascular adverse events (MACE)’ – please revise to standard work sequence to match the acronym

- Response: Attending the reviewer’s suggestion, we have revised to standard work sequence to match the acronym in the Introduction (page 3, paragraph 1, line 7): “In this particular scenario, a complete revascularization (CR) strategy of the nonculprit lesions has been associated with a reduction in major adverse cardiovascular events (MACE)4-8”.

‘…(such as increasing the stress of the patients,…’ - I would consider removing this if you do not have references to support. In my opinion some patients may be stressed to wait in hospital while others may to stressed to have lesions left unmanaged at discharged. Accordingly, best to just remove this statement.

- Response: Attending the reviewer’s comment, we have removed this statement from the Introduction (page 3, paragraph 1, line 11): “However, this non-delayed strategy may increase the risk of side effects (such as contrast induced nephropathy or bleeding)” . 

‘Thus, in this study, we aimed to examine the effect of both CR trategies on the length of stay of index hospitalization in STEMI patients with MVD.’ Since your hypothesis is also that this will be completed safely, I would consider revising to read something like this – if you agree “”Thus, in this study, we aimed to examine the effect of both CR strategies on the length of stay of index hospitalization in STEMI patients with MVD and secondarily their MACE events.””

-Response: Attending the reviewer’s suggestion, we have added this sentence to the Introduction (page 3, paragraph 1, line 21): “ Thus, in this study, we aimed to examine the effect of both CR strategies on the length of stay of index hospitalization in STEMI patients with MVD and secondarily their MACE events”.

Methods

‘For defining ischemia-driven revascularization, we included all PCI or CABG occurring after the baseline procedure and justified by recurrent symptoms or objective evidence of significant ischemia on provocative testing.’ Please clarify this statement. I assume that it means all revascularization after randomization and not after the index procedure? It would be atypical to count events prior to randomization.

- Response: Attending the reviewer’s comment, we have changed this sentence in the Methods (page 5, paragraph 4, line 6): “ For defining ischemia-driven revascularization, we included all PCI or CABG occurring after randomization and justified by recurrent symptoms or objective evidence of significant ischemia on provocative testing”.

Results

Abstract ‘At 12-month of follow-up, no differences were found in the occurrence of MACE, 7 (5.34%) patients in in-hospital CR and 4 (3.15%) in after-discharge CR strategy; (hazard ratio, 0.59; 95% confidence interval, 0.17 to 2.02; p=0.397).’

Table 10 (7.63) vs. 6 (4.72) 0.59 (0.17- 2.02)’

Results text section ‘At 12-month of follow-up, the secondary endpoint occurred in 7 of 131 patients (5.34%) in the in-hospital CR strategy and in 4 of 127 (3.15%).

The key secondary results appear different in the abstract, the results table and the results text?

- Response: Attending the reviewer’s comment, we have corrected the error. Due to an error, in Table 3, in the secondary outcome paragraph, the total number of events were expressed incorrectly. We counted separately the patients with all cause death, myocardial infarction and ischemia-driven revascularization, whilst the patients with myocardial infarction and ischemia-driven revascularization are really the same patients. That is why in the composite of the secondary outcome the real final events are 7 (5.34%) in the in-hospital group and 4 (3.15%) in the after-discharge group, and are the same as in the Abstract and the results text. Therefore, we have changed the results in Table 3(page 9): “Secondary outcome: 7(5.34) vs. 4 (3.15) 0.59 (0.17-2.02) p=0.397”.

Discussion

‘Although an immediate, during primary PCI, nonculprit PCI strategy has not been explored, we believe that it would not lead to a further reduction in the length of stay.’ I would revise this since there are several trials that have explored immediate culprit plus non-culprit PCI in the setting of MVD and STEMI.

- Response: In accordance with the reviewer, there are several trials that have explored immediate culprit plus non-culprit PCI in the setting of MVD and STEMI. What we wanted to say in that sentence, was that “Although an immediate, during primary PCI, nonculprit PCI strategy has not been explored in our work…”. That is why, in order to avoid misunderstandings and attending the reviewer’s comment, we have added to the text the next sentence (page 11, paragraph 2, line 2): “Although an immediate, during primary PCI, nonculprit PCI strategy has not been explored in our work, we believe that it would not lead to a further reduction in the length of stay”.

Additional discussion comments – the discussion is a bit repetitive that the current work is novel. Although I agree it is important and novel – this does not need to be stated more that once.

- Response: Attending the reviewer’s comment, we have only left this initial sentence in the Discussion (page 10, paragraph 4, lines 1-3): “To the best of our knowledge, this is the first RCT to evaluate the impact of different revascularization strategies on the length of hospital stay in patients with STEMI and MVD”.

 

- Response: We ensure that our manuscript meets PLOS ONE’s style requirements.

2. We note that you have selected “Clinical Trial” as your article type. PLOS ONE requires that all clinical trials are registered in an appropriate registry (the WHO list of approved registries is at https://www.who.int/clinical-trials-registry-platform/network/primary-registries" https://www.who.int/clinical-trials-registry-platform/network/primary-registries and more information on trial registration is at http://www.icmje.org/about-icmje/faqs/clinical-trials-registration/). Please state the name of the registry and the registration number (e.g. ISRCTN or ClinicalTrials.gov) in the submission data and on the title page of your manuscript. a) Please provide the complete date range for participant recruitment and follow-up in the methods section of your manuscript. b) If you have not yet registered your trial in an appropriate registry, we now require you to do so and will need confirmation of the trial registry number before we can pass your paper to the next stage of review. Please include in the Methods section of your paper your reasons for not registering this study before enrolment of participants started. Please confirm that all related trials are registered by stating: “The authors confirm that all ongoing and related trials for this drug/intervention are registered”. Please see http://journals.plos.org/plosone/s/submission-guidelines#loc-clinical-trialsfor our policies on clinical trials.

- Response: We have stated the name of the registry and the registration number in the title page (Page 1): “ClinicalTrials.gov number: NCT04743154”.

- Response: We have stated the following financial disclosures (page 13): “Funding: None to declare. The funders had no role in study design, data collection and analysis, decision to publish, or preparation of the manuscript. The authors received no specific funding for this work”.

- Response: We specify that our study’s minimal data set underlying the results described in our manuscript has been uploaded and can be found as a Supporting Information file.

---

## [Decision Letter · Decision Letter 1]

5 Mar 2024

PONE-D-23-18808R1In-hospital versus after-discharge complete revascularization in patients with ST segment elevation myocardial infarction and multivessel disease. REVIVA-ST trial.PLOS ONE

Dear Dr. Rumiz,

Thank you for submitting your manuscript to PLOS ONE. After careful consideration, we feel that it has merit but does not fully meet PLOS ONE’s publication criteria as it currently stands. Therefore, we invite you to submit a revised version of the manuscript that addresses the points raised during the review process.

We look forward to receiving your revised manuscript.

Kind regards,

Giuseppe Gargiulo, MD, PhD

Academic Editor

PLOS ONE

Additional Editor Comments:

The authors have addressed most of the comments but there is still much room for improvements.

I kindly recommend the authors to revise the manuscript according to reviewers' comments and to update the introduction and discussion on the topic by citing most recent trials, international guidelines and reviews on the topic (such as PMID: 33036712; PMID: 37634190; PMID: 37622654; PMID: 38343368)

Reviewers' comments:

Reviewer's Responses to Questions

**Comments to the Author**

1. If the authors have adequately addressed your comments raised in a previous round of review and you feel that this manuscript is now acceptable for publication, you may indicate that here to bypass the “Comments to the Author” section, enter your conflict of interest statement in the “Confidential to Editor” section, and submit your "Accept" recommendation.

Reviewer #1: All comments have been addressed

Reviewer #3: (No Response)

Reviewer #4: (No Response)

2. Is the manuscript technically sound, and do the data support the conclusions?

Reviewer #1: Yes

Reviewer #3: Partly

Reviewer #4: Partly

3. Has the statistical analysis been performed appropriately and rigorously? 

Reviewer #1: Yes

Reviewer #3: Yes

Reviewer #4: Yes

4. Have the authors made all data underlying the findings in their manuscript fully available?

Reviewer #1: Yes

Reviewer #3: (No Response)

Reviewer #4: Yes

5. Is the manuscript presented in an intelligible fashion and written in standard English?

Reviewer #1: Yes

Reviewer #3: Yes

Reviewer #4: Yes

6. Review Comments to the Author

Reviewer #1: The colleagues revised the manuscript according to our revision, and the quality of the manuscript is improved.

However, there are other issue to solve:

-There are still typos in the text. In the discussion section for example there is still a “de”. Please be sure to revise all typos.

“However,inthisstudyonly40%ofpatientspresentedwithSTEMIandthe

staged PCI could be performed within 6 weeks after de index PCI, both during the hospitalization or after discharge. Moreover, we do not actually know whether the increase in hospital stay is a consequence of a longer index admission or secondary to the deferred procedure.

-In the sentence “The current guidelines recommendations on revascularization of non-culprit lesions in STEMI , is based on the data from RCTs that evaluate the benefit of a CR over culprit vessel only PCI. However, as far as we know, there are actually no RCTs focusing on the efficacy of different CR strategies in STEMI patients. The BIOVASC trial15 showed that an inmediante CR was non-inferior to a staged CR for the combined endpoint (all cause mortality, MI, unplanned ischaemia- driven revascularization or cerebrovascular event) at 12 months of follow-up. However, this trial was performed in a cohort of patients with ACS, where 60% were non-STEMI patients.”

First of all, the sentence “The current guidelines recommendations on revascularization of non-culprit lesions in STEMI , is based on the data from RCTs that evaluate the benefit of a CR over culprit vessel only PCI” must be revised with “The current guidelines recommendations on revascularization of non-culprit lesions in STEMI , are based on the data from RCTs that evaluate the benefit of a CR over culprit vessel only PCI.” Also, there is another typo in the text (“inmediante”).

Finally, the sentence “However, as far as we know, there are actually no RCTs focusing on the efficacy of different CR strategies in STEMI patients.” is not true. A subgroup analysis of the CVLPRIT trial as well as Corpus et al ( Corpus RA, House JA, Marso SP, et al. Multivessel percutaneous coronary intervention in patients with multivessel disease and acute myocardial infarction. Am Heart J. 2004;148(3):493-500. doi:10.1016/j.ahj.2004.03.051) compared different strategies of revascularization. Please revise.

Reviewer #3: The Authors submitted the manuscript "In-hospital versus after-discharge complete revascularization in patients with ST segment elevation myocardial infarction and multivessel disease. REVIVA-ST trial."

A RCT comparing in-hospital vs discharge coronary revascularization in patients with STEMI.

Some comments:

1) Please refer to the last ESC ACS guidelines of 2023

2) The Authors should add more info on the identification of events of interest during follow-up. How were identified? Any remote follow-up beyond the outpatient visits?

3) What about contrast medium? Any difference between group? The Authors should provide info on eGFR/creatinine at baseline and 12-month follow-up to have further info on safety profile between the 2 revascularization strategies.

4) The Authors should include peri-procedural bleeding events. Did they find any differences between groups?

5) I suggest to add info on vascular access complications for completeness of safety profile comparing the two strategies.

Reviewer #4: The manuscript addresses an interesting topic. The employed statistical methods are rather sound, though standard. A few more details are required to ensure the reliability of the results. Comments follow.

1. Strict assumptions must be met to get reliable inference when t-tests are employed. Please, provide evidence that all the assumptions, from the normality to homoskedasticity, etc. are met.

2.As is the case for a linear or generalized linear model, it is desirable to determine whether a fitted Cox regression model adequately describes the data. I will briefly consider three kinds of diagnostics: for violation of the assumption of proportional hazards; for influential data; and for nonlinearity in the relationship between the log hazard and the covariates. All of these diagnostics use the residuals method, which requires the calculation of several kinds of residuals (along with some quantities that are not normally thought of as residuals). Tests and graphical diagnostics for proportional hazards may be based on the scaled Schoenfeld residuals. Nonlinearity – that is, an incorrectly specified functional form in the parametric part of the model – is a potential problem in Cox regression. The martingale residuals may be plotted against covariates to detect nonlinearity, and may

also be used to form component-plus-residual (or partial-residual) plots, again in the manner of linear and

generalized linear models.

3. I strongly suggest to provide more information on the relevant variables in the survival model. Maybe, a lasso approach could be used to select the relevant variables, dropping redundant and non-informative variables.

7. PLOS authors have the option to publish the peer review history of their article (what does this mean?). If published, this will include your full peer review and any attached files.

Reviewer #1: No

Reviewer #3: No

Reviewer #4: No

---

## [Author Response · Author response to Decision Letter 1]

18 Apr 2024

We acknowledge to the Editorial Office and Reviewers for their comments and constructive suggestions, which undoubtedly have improved the quality of this manuscript.

Additional Editor Comments:

The authors have addressed most of the comments but there is still much room for improvements.

I kindly recommend the authors to revise the manuscript according to reviewers' comments and to update the introduction and discussion on the topic by citing most recent trials, international guidelines and reviews on the topic (such as PMID: 33036712; PMID: 37634190; PMID: 37622654; PMID: 38343368).

Response: Following the editor’s comments, we updated the introduction and discussion on the topic by citing the recommended recent trials, international guidelines and reviews on the topic.

- Discussion, page 11: “Multiple studies have evaluated the efficacy of a CR in STEMI patients with MVD, however, even today, the optimal time to perform it still remains a matter of debate14. In this regard, our study demonstrates…”

- References, page 16: “14. Ilardi F, Ferrone M, Avvedimento M, Servillo G, Gargiulo G. Complete Revascularization in Acute and Chronic Coronary Syndrome. Cardiol Clin. 2020;38(4):491-505.”

- Discussion, pages 11 and 12: “Although an immediate, during primary PCI, nonculprit PCI strategy has not been explored in our work, we believe that it would not lead to a further reduction in the length of stay. Recently, the Multivessel Immediate versus Staged Revascularization in Acute Myocardial Infarction (MULTISTARS AMI) trial, demonstrated that immediate multivessel PCI at the time of primary PCI was noninferior to staged multivessel PCI in patients with STEMI and MVD, reporting a median hospital stay of 4 days (3-6) in the immediate CR strategy, similar to the data obtained in the delayed CR group of our work17. Likewise, Politi et al observed no differences in this setting between STEMI patients revascularized during primary PCI or in a staged procedure17. By contrast,”

- Discussion, page 13: ¨ Accordingly, the MULTISTARS trial also presented an increase in 12-month MACE, mostly due to nonfatal MI and unplanned ischemia-driven revascularization, in the staged CR strategy. It is important to highlight that, no apparent differences in the percentage of patients with spontaneous MI were observed between the trial groups. Furthermore, the difference observed was predominantly due to an elevated rate of procedure-related MI, which is a really difficult MACE to quantify in an immediate strategy.”

- References, page 16: “16. Stähli BE, Varbella F, Linke A, Schwarz B, Felix SB, Seiffert M, et al. Timing of Complete Revascularization with Multivessel PCI for Myocardial Infarction. N Engl J Med. 2023;389(15):1368-1379.”

- Introduction, page 3: “but always within the first 45 days. Therefore, the recent 2023 ESC Guidelines for the management of acute coronary syndromes, recommend a CR either during the index PCI procedure or within 45 days (class of recommendation I, level of evidence A)10. Since the optimal timing of revascularization (immediate during the index procedure vs. staged within 45 days) has not yet been evaluated in adequate randomized clinical trials (RCTs), no recommendation has been endorsed in this sense in the guidelines.¨

- References, page 16: “10. Byrne RA, Rossello X, Coughlan JJ, Barbato E, Berry C, Chieffo A, et al. 2023 ESC Guidelines for the management of acute coronary syndromes. Eur Heart J. 2023; 44(38):3720–3826.”

- Introduction, page 3: “Both of these strategies have pros and cons11, and therefore, the optimal timing for CR still today remains a subject of debate.”

- References, page 16: “11. Kastrati A, Kessler T, Rinaldi R, Brugaletta S. Complete revascularisation should be immediate in STEMI: pros and cons. EuroIntervention. 2024;20(3):e171-e173.”

Review Comments to the Author

Reviewer #1: The colleagues revised the manuscript according to our revision, and the quality of the manuscript is improved.

However, there are other issue to solve:

-There are still typos in the text. In the discussion section for example there is still a “de”. Please be sure to revise all typos.

“However, in this study only 40% of patients presented with STEMI and the staged PCI could be performed within 6 weeks after de index PCI, both during the hospitalization or after discharge. Moreover, we do not actually know whether the increase in hospital stay is a consequence of a longer index admission or secondary to the deferred procedure”.

Response: In accordance with the reviewer’s comment, we have revised all typos in the text.

- Discussion, page 11: “However, in this study only 40% of patients presented with STEMI and the staged PCI could be performed within 6 weeks after the index PCI, both during the hospitalization or after discharge”.

-In the sentence “The current guidelines recommendations on revascularization of non-culprit lesions in STEMI, is based on the data from RCTs that evaluate the benefit of a CR over culprit vessel only PCI. However, as far as we know, there are actually no RCTs focusing on the efficacy of different CR strategies in STEMI patients. The BIOVASC trial15 showed that an inmediante CR was non-inferior to a staged CR for the combined endpoint (all cause mortality, MI, unplanned ischaemia- driven revascularization or cerebrovascular event) at 12 months of follow-up. However, this trial was performed in a cohort of patients with ACS, where 60% were non-STEMI patients.”

First of all, the sentence “The current guidelines recommendations on revascularization of non-culprit lesions in STEMI, is based on the data from RCTs that evaluate the benefit of a CR over culprit vessel only PCI” must be revised with “The current guidelines recommendations on revascularization of non-culprit lesions in STEMI , are based on the data from RCTs that evaluate the benefit of a CR over culprit vessel only PCI.” Also, there is another typo in the text (“inmediante”).

Finally, the sentence “However, as far as we know, there are actually no RCTs focusing on the efficacy of different CR strategies in STEMI patients.” is not true. A subgroup analysis of the CVLPRIT trial as well as Corpus et al (Corpus RA, House JA, Marso SP, et al. Multivessel percutaneous coronary intervention in patients with multivessel disease and acute myocardial infarction. Am Heart J. 2004;148(3):493-500. doi:10.1016/j.ahj.2004.03.051) compared different strategies of revascularization. Please revise.

Response: Following the reviewer’s suggestion, we have modified the following sentences in the text.

- Discussion, page 12: “The current guidelines recommendations on revascularization of nonculprit lesions in STEMI9, are based on the data from RCTs that evaluate the benefit of a CR over culprit vessel only PCI. However, as far as we know, actually exists scarce scientific evidence that focuses on the evaluation of the optimal-timing of the different CR strategies in STEMI patients with MVD. The BIOVASC trial16 showed that an immediate CR was non-inferior to a staged CR for the combined endpoint (all cause mortality, MI, unplanned ischaemia-driven revascularization or cerebrovascular event) at 12 months of follow-up. However, this trial was performed in a cohort of patients with ACS, where 60% were non-STEMI patients.”

Reviewer #3: The Authors submitted the manuscript "In-hospital versus after-discharge complete revascularization in patients with ST segment elevation myocardial infarction and multivessel disease. REVIVA-ST trial."

A RCT comparing in-hospital vs discharge coronary revascularization in patients with STEMI.

Some comments:

1) Please refer to the last ESC ACS guidelines of 2023.

Response: Following the reviewer’s comment, we have added the recent 2023 ESC ACS Guidelines in the references and we have modified the text in order to refer to these guidelines.

- Introduction, page 3: “However, the optimal timing to perform this “protective” PCI remains uncertain. The 2017 European Society of Cardiology (ESC) STEMI Guidelines recommended a CR strategy preferably during index admission before discharge9. The main rationale for this recommendation was due to the fact that all trials available until then had performed MVD PCI within that specific time frame. However, this non-delayed strategy may increase the risk of side effects (such as contrast induced nephropathy or bleeding) and logistic constrains that ultimately increase the length of index hospitalization. Recently, the COMPLETE trial, demonstrated a 3-year reduction in the combined endpoint of death or MI with staged PCI (performed within 45 days of STEMI) of the non-infarct related artery compared with culprit-vessel only PCI8. These benefits were consistent, irrespective of the timing of the non-infarct artery PCI, whether it was performed during admission or after discharge, but always within the first 45 days. Therefore, the recent 2023 ESC Guidelines for the management of acute coronary syndromes, recommend a CR either during the index PCI procedure or within 45 days (class of recommendation I, level of evidence A)10. Since the optimal timing of revascularization (immediate during index hospitalization vs. staged within 45 days) has not yet been evaluated in adequate randomized clinical trials (RCTs), no recommendation has been endorsed in this sense in these guidelines. Along with these lines of thoughts, we speculate that…”

- References, page 16: “10. Byrne RA, Rossello X, Coughlan JJ, Barbato E, Berry C, Chieffo A, et al. 2023 ESC Guidelines for the management of acute coronary syndromes. Eur Heart J. 2023; 44(38):3720–3826.”

2) The Authors should add more info on the identification of events of interest during follow-up. How were identified? Any remote follow-up beyond the outpatient visits?

Response: In accordance with the reviewer’s comment, we have to clarify that the identification of events of interest during follow-up were not only identified during the scheduled outpatient visits, but also by interviewing the hospital electronic database at 1 year (�14 days) after index revascularization. Therefore, we have modified the following paragraph:

- Methods, page 6: “Follow-up was conducted during outpatient clinic visits scheduled at 30 days, 6 months and 12 months after primary revascularization, and also by means of hospital electronic database.”

3) What about contrast medium? Any difference between group? The Authors should provide info on eGFR/creatinine at baseline and 12-month follow-up to have further info on safety profile between the 2 revascularization strategies.

Response: According to the reviewer`s suggestion, we added the volume of contrast used, as well as the creatinine at baseline and at 12-month follow-up in table 1 and 2.

- Results, Table 1, page 8: 

Creatinine at baseline, mg/dl 1.5 � 0.55 0.98 � 0.75 0.862

- Results, Table 2, page 8 and 9:

Volume of contrast used, ml 110 (90-115) 105 (72-108) 0.746

Creatinine at 12-month follow-up, mg/dl 1.48 � 0.43 0.96 � 0.61 0.634

4) The Authors should include peri-procedural bleeding events. Did they find any differences between groups?

Response: Following the reviewer`s suggestion, we added periprocedural major bleedings in table 2, observing no differences between both groups. Therefore, we added the following sentence to the text:

- Results, page 7: “No differences were observed between the two group strategies in terms of peri-procedural major bleeding (BARC � 3) or vascular access complications.”

- Results, Table 2, page 9: 

Peri-procedural major bleeding, n (%) 2 (1.53) 0 (0) 0.161

5) I suggest to add info on vascular access complications for completeness of safety profile comparing the two strategies. 

Response: In accordance with the reviewer`s comment, we added vascular access complications in table 2, observing no statistical differences between both group strategies. Therefore, we added the following sentence to the text:

- Results, page 7: “No differences were observed between the two group strategies in terms of peri-procedural major bleeding (BARC � 3) or vascular access complications.”

- Results, Table 2, page 9: 

Vascular access complication, n (%) 1 (0.76) 0 (0) 0.320

Reviewer #4: The manuscript addresses an interesting topic. The employed statistical methods are rather sound, though standard. A few more details are required to ensure the reliability of the results. Comments follow.

1. Strict assumptions must be met to get reliable inference when t-tests are employed. Please, provide evidence that all the assumptions, from the normality to homoskedasticity, etc. are met.

Response: You are right. The results from the Shapiro-Wilk test for length of stays indicate that the data do not follow a normal distribution, as shown below.

W: The W value is 0.85360. This is the test statistic and reflects how well the data fit a normal distribution. Values closer to 1 indicate a better fit to the normal distribution.

Prob>z: The p-value is 0.00000, which is less than the common significance threshold of 0.05 (or any other threshold you might be using, such as 0.01). This means there is sufficient statistical evidence to reject the null hypothesis that the data are normally distributed.

In the revised version of the text, we reanalyze the between-treatment difference for length of stay using the Mann-Whitney test. This analysis revealed the median length of stays was lower in the active arm (as shown below). The methods section were changed accordingly.

- Methods, page 6: All patients who underwent randomization were included in the analysis according to the treatment group to which they were assigned, regardless of the CR strategy they actually received (intention-to-treat principle). Because the length of stay did not meet the criteria for a parametric distribution (assessed by the Shapiro-Wilk test), the primary endpoint (length of stay) between treatment arms was analyzed by the Mann-Whitney test. 

Accordingly, the results section was changed. Basically, the meaning are in the same line. Patients in the active arm showed a lower length of stay. 

- Results, page 9: Clinical outcomes are summarized in Table 3. The median length of hospital stay (p25%-p75%) was significantly higher in the in-hospital CR strategy, when compared with the after-discharge CR group [7 days (5-9) versus 4 days (3-5); p<0.001]. There were no differences between both strategies in terms of length of stay in the median coronary care unit (CCU) [2 days (2-3) vs 2 days (2-3), p=0.631], respectively. The median length of stay after a non-culprit revascularization procedure in the after-discharge strategy was inferior to 24 hours in all patients [0.38 days (0.25-0.81)]. 

2.As is the case for a linear or generalized linear model, it is desirable to determine whether a fitted Cox regression model adequately describes the data. I will briefly consider three kinds of diagnostics: for violation of the assumption of proportional hazards; for influential data; and for nonlinearity in the relationship between the log hazard and the covariates. All of these diagnostics use the residuals method, which requires the calculation of several kinds of residuals (along with some quantities that are not normally thought of as residuals). Tests and graphical diagnostics for proportional hazards may be based on the scaled Schoenfeld residuals. Nonlinearity – that is, an incorrectly specified functional form in the parametric part of the model – is a potential problem in Cox regression. The martingale residuals may be plotted against covariates to detect nonlinearity, and may also be used to form component-plus-residual (or partial-residual) plots, again in the manner of linear and generalized linear models.

Response: Thanks for your input. Following your suggestion, we tested the proportionality assumption of the Cox regression model using Schoenfeld residuals. 

Revcompext (type of revascularization strategy)

As you may appreciate in the output (see below) the hazard ratios between groups are constant over time. The test for this assumption, based on the rank of analysis time and focusing on the revcompext variable (our exposure), gives a rho value of -0.11517 

---

## [Decision Letter · Decision Letter 2]

24 Apr 2024

In-hospital versus after-discharge complete revascularization in patients with ST segment elevation myocardial infarction and multivessel disease. REVIVA-ST trial.

PONE-D-23-18808R2

Dear Dr. Rumiz,

We’re pleased to inform you that your manuscript has been judged scientifically suitable for publication and will be formally accepted for publication once it meets all outstanding technical requirements.

Kind regards,

Giuseppe Gargiulo, MD, PhD

Academic Editor

PLOS ONE

Additional Editor Comments (optional):

Reviewers' comments:

Reviewer's Responses to Questions

**Comments to the Author**

1. If the authors have adequately addressed your comments raised in a previous round of review and you feel that this manuscript is now acceptable for publication, you may indicate that here to bypass the “Comments to the Author” section, enter your conflict of interest statement in the “Confidential to Editor” section, and submit your "Accept" recommendation.

Reviewer #3: All comments have been addressed

2. Is the manuscript technically sound, and do the data support the conclusions?

Reviewer #3: Yes

3. Has the statistical analysis been performed appropriately and rigorously? 

Reviewer #3: I Don't Know

4. Have the authors made all data underlying the findings in their manuscript fully available?

Reviewer #3: Yes

5. Is the manuscript presented in an intelligible fashion and written in standard English?

Reviewer #3: Yes

6. Review Comments to the Author

Reviewer #3: The Authors addressed all of my questions and I have no further comments to the manuscript. Best regards.

7. PLOS authors have the option to publish the peer review history of their article (what does this mean?). If published, this will include your full peer review and any attached files.

Reviewer #3: No

---

## [Editor Report · Acceptance letter]

30 Apr 2024

PONE-D-23-18808R2 

PLOS ONE

Dear Dr. Rumiz, 

I'm pleased to inform you that your manuscript has been deemed suitable for publication in PLOS ONE. Congratulations! Your manuscript is now being handed over to our production team.

Kind regards, 

on behalf of

Dr. Giuseppe Gargiulo 

Academic Editor

PLOS ONE